# Leukaemia Inhibitory Factor (LIF) Inhibits Cancer Stem Cells Tumorigenic Properties through Hippo Kinases Activation in Gastric Cancer

**DOI:** 10.3390/cancers12082011

**Published:** 2020-07-22

**Authors:** Lornella Seeneevassen, Julie Giraud, Silvia Molina-Castro, Elodie Sifré, Camille Tiffon, Clémentine Beauvoit, Cathy Staedel, Francis Mégraud, Philippe Lehours, Océane C.B. Martin, Hélène Boeuf, Pierre Dubus, Christine Varon

**Affiliations:** 1INSERM U1053, Bordeaux Research in Translational Oncology, Univ. Bordeaux, F-33000 Bordeaux, France; lornella.seeneevassen@u-bordeaux.fr (L.S.); julie.giraud085@gmail.com (J.G.); smolina13@gmail.com (S.M.-C.); elodie.sifre@u-bordeaux.fr (E.S.); camille.tiffon@u-bordeaux.fr (C.T.); clementine.beauvoit@etu.u-bordeaux.fr (C.B.); francis.megraud@chu-bordeaux.fr (F.M.); philippe.lehours@u-bordeaux.fr (P.L.); oceane.martin@u-bordeaux.fr (O.C.B.M.); pierre.dubus@u-bordeaux.fr (P.D.); 2Institute for Health Research, University of Costa Rica, San José 11502, Costa Rica; 3INSERM U1212, Régulations Naturelles et Artificielles des ARNs, Univ. Bordeaux, F-33000 Bordeaux, France; Cathy.staedel@inserm.fr; 4Centre National de Référence des Helicobacters et Campylobacters, CHU Bordeaux, F-33000 Bordeaux, France; 5INSERM U1026, Bioingénierie Tissulaire, Univ. Bordeaux, F-33000 Bordeaux, France; helene.boeuf@u-bordeaux.fr; 6Department of Histology and Pathology, CHU Bordeaux, F-33000 Bordeaux, France

**Keywords:** gastric carcinoma, GP190, LATS1/2, YAP, CD44, ALDH, JAK, Ruxolitinib, XMU-MP-1

## Abstract

Cancer stem cells (CSCs) present chemo-resistance mechanisms contributing to tumour maintenance and recurrence, making their targeting of utmost importance in gastric cancer (GC) therapy. The Hippo pathway has been implicated in gastric CSC properties and was shown to be regulated by leukaemia inhibitory factor receptor (LIFR) and its ligand LIF in breast cancer. This study aimed to determine LIF’s effect on CSC properties in GC cell lines and patient-derived xenograft (PDX) cells, which remains unexplored. LIF’s treatment effect on CSC markers expression and tumoursphere formation was evaluated. The Hippo kinase inhibitor XMU-MP-1 and/or the JAK1 inhibitor Ruxolitinib were used to determine Hippo and canonical JAK/STAT pathway involvement in gastric CSCs’ response to LIF. Results indicate that LIF decreased tumorigenic and chemo-resistant CSCs, in both GC cell lines and PDX cells. In addition, LIF increased activation of LATS1/2 Hippo kinases, thereby decreasing downstream YAP/TAZ nuclear accumulation and TEAD transcriptional activity. LIF’s anti-CSC effect was reversed by XMU-MP-1 but not by Ruxolitinib treatment, highlighting the opposite effects of these two pathways downstream LIFR. In conclusion, LIF displays anti-CSC properties in GC, through Hippo kinases activation, and could in fine constitute a new CSCs-targeting strategy to help decrease relapse cases and bad prognosis in GC.

## 1. Introduction

Gastric cancer is the fifth most common cancer and the third leading cause of cancer-related deaths worldwide with about 1,033,701 new cases and 782,685 deaths in 2018, in both sexes [1]. Most cases are gastric adenocarcinomas (GCs) which vary histologically and can be classified into three subtypes according to the Lauren classification: the intestinal, diffuse and mixed subtypes. GCs are usually detected late, most of the time at the metastatic stage. Treatment consists of surgery with additional chemo/radiotherapies, but the number of relapse cases remains high with a five-year survival rate less than 20%; less than 5% for advanced unresectable or metastatic cases which account for about 80% patients at diagnosis [2]. There exists at present no targeted therapy except for Herceptin which can be used to treat HER2+ GC cases comprising only 30% of GCs [3].

Studies are more and more incriminating cancer stem cells (CSCs) in GC bad prognosis. Experiments using different human GC cell lines and mouse models of patient-derived xenograft tumours (PDX) have demonstrated the role of this small cell subpopulation in GC initiation, growth, chemo-resistance, relapse as well as metastasis [4,5]. These cells possess unlimited self-renewal characteristics as well as asymmetric division and differentiation properties, creating the hierarchical structure of tumours. CSCs are involved in tumour recurrence due to their highly chemo-resistant properties and targeting them could help limit tumour resistance and relapse.

One characteristic of cancer cells and CSCs is the hijack of cell signalling pathways for their survival and maintenance [6,7]. Accumulating data suggest the disruption of the Hippo signalling pathway in many human cancers including GC, which often correlates with poor patient prognosis [8]. This pathway is made up of two distinct modules, a regulatory tumour suppressor kinase core composed of MST1/2 upstream LATS1/2 and a transcriptional module with the oncoproteins YAP and TAZ. The Hippo pathway and its effectors YAP/TAZ are tightly regulated physiologically in respect to their role in the maintenance of the highly ordered architecture of tissues and organs [9]. YAP/TAZ nuclear translocation and interaction with transcriptional factors from the TEAD family are responsible for the main physiological effects of the Hippo pathway in organ size during development and stemness [10]. MST1/2-phosphorylated-LATS1/2 directly phosphorylates YAP/TAZ, causing either their cytoplasmic retention or proteasomal degradation, thus regulating their activity. Dysregulation of this pathway in cancer increases the ability of YAP/TAZ oncoproteins to translocate to the nucleus and act as co-factors, inducing the expression of a core of oncogenic genes: cell-proliferative and anti-apoptotic genes, for instance [8]. The Hippo oncogenic effectors YAP/TAZ are abnormally detected in the nucleus of tumour cells in tumorous tissue where they induce growth-promoting transcriptional programs [8]. Transgenic expression of YAP in mouse liver leads to cell transformation and tumour development, and induced expression of YAP can trigger CSC properties [11]. Evidence also links the Hippo pathway to breast CSCs with YAP being overactivated in poorly differentiated breast cancers (G3) which are enriched in CSCs, compared with well-differentiated breast cancers (G1) [12].

In this respect, our team recently showed the role of the Hippo pathway in gastric carcinogenesis and gastric CSC maintenance, making this pathway a potential candidate for anti-CSC targeted therapies [13,14]. Using models previously developed in our laboratory for the study of gastric CSCs and markers like CD44, which can enrich cells with CSC properties [5,15], we demonstrated that targeting Hippo effectors through silencing RNA strategies decreased the gastric CSC population and properties [14]. Moreover, Verteporfin, an FDA-approved drug used for the treatment of age-related macular degeneration, was repositioned as a YAP/TEAD inhibitor, and demonstrated anti-tumorigenic effects via the inhibition of gastric CSC properties both in vitro and in vivo [14].

Hippo pathway regulators are diverse, from inter-cellular junctions and cell polarity, to cell surface receptors [9]. Chen et al. showed that the leukaemia inhibitory factor receptor (LIFR) acts upstream the Hippo pathway, limiting oncogenic YAP transfer to the nucleus, and is downregulated in human breast cancer, which correlates with poor clinical conditions [16,17]. Interestingly, treatment with LIFR ligand leukaemia inhibitory factor (LIF) and ectopic LIFR expression act as metastasis suppressors in breast cancer [16].

LIF, a member of the IL-6 family of cytokines, displays pleiotropic effects depending on cell types and organs. First cloned as an inducer of differentiation and inhibitor of proliferation of the myeloid leukemic cell line M1 [18], LIF also presented pro-proliferation effects on adult human T cells [19] as well as anti-differentiation properties on murine embryonic stem (ES) cells [20]. It has a pivotal role in embryo implantation and uterine receptivity regulation [21], in the maintenance of hematopoietic stem cell pools, in muscle regeneration as well as in neural injury regulation, amongst others [22]. The past few years have seen an uprising interest for LIF effects in cancer. Apart from its anti-metastatic effect in breast cancer [16,17], recent studies showed that LIF/LIFR signalling could negatively regulate metastasis of pancreatic cancer (PC) and hepatocellular carcinoma (HCC) cells both in vitro and in vivo [23,24]. Conversely, and in relation to its pleiotropic effects, high LIFR expression was found to promote melanoma cell migration and unfavourable prognosis for melanoma patients [25] as well as aggressiveness of chordomas [26].

Interestingly, the role of LIF/LIFR signalling in the gastric cancer context has not yet been thoroughly investigated. Indeed, there are only few articles describing LIF/LIFR effects in gastric cancer and none of them addressed CSCs. Zhao et al. showed that the long noncoding RNA LOWEG, which is under-expressed in GC tissues and cell lines compared with non-tumorous gastric mucosa, is a tumour suppressor that can decrease GC tumorigenicity and positively affect LIF/LIFR signalling when overexpressed in GC cells [27]. In addition, Xu et al. recently showed that LIF overexpression as well as recombinant LIF treatment inhibit GC proliferation by inducing G1-phase arrest and delayed tumour formation in vivo [28].

Despite the interesting anti-tumoral effect of LIF/LIFR suggested by these studies, the cellular signalling mechanisms behind these effects have not been explored. Indeed, LIF pleiotropic effects can be attributed to its capacity to solicit different cell signalling pathways downstream its receptor. LIF binding on GP190 (LIFRβ) subunit triggers the recruitment of GP130 subunit and heterodimerisation, leading to Janus kinase (JAK) 1 phosphorylation followed by the activation of several cell signalling pathways [29]. The JAK and its transcriptional factor, Signal Transducer and Activator of Transcription (STAT), pathway is the most studied cell signalling pathway in the LIF/LIFR context. JAK1 phosphorylation induces STAT3 phosphorylation, dimerisation and nuclear translocation for the transcription of genes involved in self-renewal, cell survival, proliferation as well as differentiation. Interestingly, LIF anti-tumorigenic effects in breast cancer [16,30] and clear cell renal carcinoma [31] were found to involve the kinase core of the Hippo pathway while LIF pro-tumorigenic effects in melanoma pass through the JAK/STAT pathway activation [25], showing how the contradictory effects of LIF in cancer depend on the cellular mechanisms involved.

In this context, the aim of this study was to decipher the effect of LIF/LIFR signalling in the GC context, more particularly on the tumorigenic properties of gastric CSCs, and to explore the cellular mechanisms involved. Using GC cell lines as well as PDX cells, we hereby showed that LIF treatment can decrease the tumoursphere-forming capacity, an important CSC characteristic. In addition, LIF decreased CSC population, revealed by a decrease in CSC markers both at the mRNA and protein levels. Interestingly, besides its canonical JAK/STAT3 signalisation, LIF activated Hippo tumour suppressor kinases LATS1/2 and inhibited YAP/TAZ-TEAD-mediated oncogenic activity by decreasing nuclear translocation of Hippo effectors YAP/TAZ. Finally, the use of the Hippo kinase inhibitor XMU-MP-1 and JAK/STAT inhibitor Ruxolitinib showed that LIF anti-CSC effects involve Hippo kinase activation by LIF and not JAK/STAT, since Hippo kinase inhibition reverted LIF-induced anti-CSC effects.

## 2. Results

### 2.1. LIF/LIFR/JAK/STAT Pathway Is Functional in GC and Gastric CSCs

To check the relevance of the LIF treatment strategy chosen for this study, GC cells’ responsiveness to LIF was verified in human GC cell lines AGS and MKN45 (Appendix A) and PDX cells GC07. Cells were supplemented with LIF at different time intervals (0–48 h or 0–24 h) and STAT3 Tyr705 phosphorylation was used as the read-out of LIF-induced activation of the LIF/LIFR canonical pathway. As expected, GC cells treatment with LIF significantly increases p-STAT3^Tyr705^ as from 30 min, especially for AGS cells (Figure 1A), it led to an increase in expression of JAK1/STAT3 target genes (Figure 1B, Appendix A).

Gastric CSCs were previously described as representing only a small proportion of GC cells [5]. JAK/STAT signature was thus checked by transcriptomic analysis on this subpopulation after CD44 based-FACS cell sorting of six different PDX-derived cells to evaluate LIF/LIFR signalisation in CD44+ gastric CSCs. Overexpression of the CSC markers CD44, ALDH1A1, CD166, CD24 and ITG6 in the CD44+ FACS-sorted cells compared with CD44− cells confirmed that the CD44 FACS-sorting was properly carried out and that the CD44+ cells were indeed CSCs (Figure 1C). CD44+ gastric PDX cells seem to present an upregulation of both JAK/STAT positive and negative regulators, showing a tightly controlled activation of this pathway in CSCs compared with non-CSCs (Figure 1C). The main transducers of the LIF/LIFR canonical JAK/STAT pathway were upregulated in CD44+ cells, including JAKs and several members of the STAT family. In addition, other JAK/STAT signalisation positive regulators like GRB2, IFNAR1 and IFNGR1 over-regulation were noted. Most JAK/STAT negative regulators, among the three major classes of inhibitors SOCS, PIAS and PTPs, were also upregulated (Figure 1C).

Those from the SOCS-family are also target genes of JAK/STAT signalling. Their expression is increased when the pathway is over-activated in order to act in turn as negative feedback regulators to retro control the pathway. In addition, the negative upregulators of the JAK/STAT pathway seemed to be more expressed than the positive regulators confirming the tight regulation of this pathway in CD44+ cells. Interestingly, LIF was significantly under-expressed in most CD44+ PDX cells analysed compared with CD44- PDX cells, strengthening the interest of LIF supplementation in GC.

Since LIF transduction implies the presence of the GP190 subunit of LIFR and since the whole GC population seems to be responsive to LIF (Figure 1A), it was important to verify the presence of LIFR-GP190 on the CSC subpopulation which would be targeted by LIF. LIFR-GP190 protein expression was examined in GC cell lines by flow cytometry. Both AGS and MKN45 cells express LIFR and interestingly, in both cell lines, CD44+ or CD44high cells, corresponding to the CSC population, expressed significantly more LIFR compared with non-CSC CD44-/low cells (Figure 1D,E). In addition, LIFR expression was not affected by LIF treatment in both CD44+/high and CD44-/low populations, suggesting that treating GC cells with LIF for 48 h does not seem to induce LIFR recycling/degrading mechanisms which might have induced non-responsiveness to LIF with time. Consequently, LIF/LIFR/JAK/STAT signal transduction observed in whole GC population after LIF treatment (Figure 1A) could be mostly attributed to that of the gastric CSC population which contains more LIFR and presents an upregulation of the JAK/STAT signature.

LIF treatment thus seems to be an appropriate strategy to target gastric CSCs since GC cells respond to LIF and CSCs show a LIF/LIFR/JAK/STAT upregulated transcriptomic signature. Besides, the LIFR-GP190 higher expression by CD44+/high cells shows that LIF/LIFR/JAK/STAT signal transduction induced after LIF treatment of a whole GC population can be attributed mostly to CSCs.

### 2.2. LIF Presents Anti-CSC Effects on GC Cell Lines and PDX Cells

LIF/LIFR signalling effects on CSC tumorigenic functional properties were then assessed after LIF treatment, through non-adherent tumoursphere-forming assays. LIF significantly decreased AGS cells’ tumourspheres-forming capacity in a dose-dependent manner (Figure 2A). The dose of 50 ng/mL LIF, which was that inducing the most important effect in AGS cells, was chosen for further experiments. Results were confirmed in the MKN45 GC cell line as well as in PDX cells GC07, GC10 and GC04 (Figure 2A), showing that LIF presents anti-tumorigenic capacities in GC cell lines as well as in cells derived from GC patients.

To confirm whether LIF’s anti-tumorigenic effect was the consequence of CSC targeting, 48-h LIF-treated cells were analysed for the expression of the CSC-associated marker CD44 by flow cytometry. The percentage of cells expressing CD44 as well as the mean expression of CD44, determined by the mean fluorescence intensity (MFI), decreased significantly in GC07 PDX cells as well as in AGS cells (Figure 2B), but not in MKN45 cells though a decrease tendency can be noted. Nguyen et al. described CD44+ALDH+ MKN45 cells as being highly tumorigenic [5]. In this regard, ALDH activity, another marker of gastric CSCs [5], was analysed by flow cytometry in MKN45 cells upon LIF treatment. All ALDH+ cells analysed were CD44+/high in accordance with Nguyen et al.’s previous data [5] (Appendix A). A significant decrease in the percentage of ALDH+ cells as well as in ALDH MFI was noted after 48-h LIF treatment (Figure 2B). These results indicate that, in MKN45 cells, even if CD44 expression remained high, ALDH activity was significantly reduced by LIF treatment. These results, combined with those observed in AGS and GC07 cells, show that LIF reduces the population of cells expressing either CD44 or ALDH CSC markers in GC cell lines and patient-derived cells.

Finally, mRNA expression of gastric CSC markers, including *CD44* and *ALDH1A1* as well as *CD24*, *CD166* and *KLF5* as previously reported [5,14,32], also diminished after LIF treatment of AGS, MKN45 and GC06 PDX cells (Figure 2C).

Altogether, these results indicate that LIF treatment induces an anti-tumorigenic effect in GC, via a decrease in CSC properties and population in both GC cell lines and PDX cells.

### 2.3. LIF/LIFR Signalling Potentiates Chemotherapy Effect on Gastric CSCs

One characteristic of CSCs, making their targeting of utmost importance in cancer therapy, is their resistance to conventional chemotherapies leading to relapse cases.

Live immunofluorescence analysis was carried out for MKN45 cells to analyse ALDH activity, coupled with CD44 protein expression and Hoescht-33342 compound incorporation, in 7-day-old spheres, treated or not with LIF for 48 h. Hoechst is a nucleic acid stain, known to be cell-permeant and to have affinity for DNA. It is incorporated by live cancer cells. However, it was precedingly shown that chemo-resistant CSCs possess efflux pumps allowing them to efflux chemotherapy drugs in cancer cases and to evacuate Hoechst in vitro [5]. This live immunofluorescence technique thus allows the analysis of the different subpopulations of CSCs contributing to tumour heterogeneity. LIF treatment decreased both ALDH+ and CD44+ cells and most particularly the ALDH+Hoechst- and CD44+Hoechst- subpopulations corresponding to chemo-resistant CSCs, as previously described [5] (Figure 3A(i)(ii)).

Conversely, an increase in ALDH-Hoechst+ and CD44-Hoescht+ cells, previously described as differentiated non-CSC and non-tumorigenic cells [5], was observed (Figure 3A(i),(ii)). These results suggest that LIF reduces the population of CSCs with drug efflux properties.

To address this point, LIF treatment was tested, in combination with conventional chemotherapeutical agents, Doxorubicin (DOX) and 5-Fluorouracil (5-FU), on GC cells’ tumoursphere-formation properties. In AGS and MKN45 cells as well as PDX cells, GC07, LIF, DOX and 5-FU alone significantly decreased the capacity of cells to form tumourspheres when compared with untreated controls (Figure 3B). Interestingly, when LIF was added in combination with DOX and 5-FU, the number of cells forming spheres in vitro decreased significantly compared with chemotherapy drugs alone (Figure 3B).

Combined, these results show that LIF, by reducing the population of chemo-resistant CSCs, potentiates the anti-tumorigenic efficiency of chemotherapy drugs in GC cell lines and also in cells derived from GC patients in vitro.

### 2.4. LIF Activates Hippo Tumour Suppressor Kinases in GC

LIF/LIFR signalling was described as being upstream the Hippo pathway in breast cancer [16]. Previous studies from our team described the Hippo pathway as being involved in gastric CSC maintenance through the tumorigenic activity of its effectors YAP/TAZ/TEAD [13,14]. In order to evaluate LIF effect on the Hippo pathway in GC, which has neither been studied nor described, Hippo pathway status and especially Hippo tumour suppressor kinase LATS1/2 activation was analysed upon LIF treatment.

Interestingly, an increase in LATS1/2^Thr1079/1041^ phosphorylation, associated with an increase in LATS1/2-mediated phosphorylation of YAP on Ser127 was found as from 30 min LIF treatment of AGS and MKN45 cells (Figure 4A). Since this LATS1/2-mediated YAP/TAZ phosphorylation leads to their cytoplasmic retention and less nuclear translocation, LIF-treated AGS and MKN45 cells were checked and quantified for nuclear YAP/TAZ. YAP and TAZ are paralogues, YAP being the main Hippo effector in AGS cells and TAZ its homologue in MKN45 cells [13,14]. Indeed, a significant decrease in nuclear YAP/TAZ was noted from 30-min LIF treatment, reflecting p-LATS1/2^Thr1079/1041^-induced phosphorylation and cytoplasmic retention of YAP/TAZ (Figure 4B,C).

In accordance with these results, kinase-induced YAP/TAZ phosphorylation decreases YAP/TAZ interaction with TEAD transcription factors assessed by TEAD-luciferase reporter assay. TEAD activity significantly decreased after LIF supplementation in both AGS and MKN45 cells, as from 30 min post-LIF for the former and 2 h for the latter (Figure 4D). To confirm the above effects, mRNA expression of genes encoding targets of this pathway was analysed (Figure 4E). Both *CYR61* and *CTGF* decreased significantly after LIF supplementation, as from 24 h for *CYR61* in AGS.

LIF thus effectively activates the Hippo pathway kinase core which in turn inhibits YAP/TAZ oncogenic effectors nuclear translocation and TEAD transcriptional activity in the GC context.

### 2.5. LIF Anti-CSC Effects Are Linked to Hippo Pathway Tumour Suppressor Core Activation

In order to address the possible link between LIF anti-CSC effects and Hippo kinase activation, the Hippo kinase MST1/2 inhibitor XMU-MP-1 was used to inhibit LATS1/2^Thr1079/1041^ phosphorylation and activation. XMU-MP-1 did not affect STAT3 and STAT3 ^Tyr705^ phosphorylation (Appendix A), while efficiently inhibiting LATS1/2^Thr1079/1041^ phosphorylation, even in the presence of LIF (Figure 5A(i)). This confirmed that the LIF-induced increase in p-LATS1/2^Thr1079/1041^ was due to MST1/2 activation mediating LATS1/2 phosphorylation on Thr1079/1041. TEAD-luciferase reporter assay carried out in the presence of XMU-MP-1 and LIF showed a significant increase in TEAD transcriptional activity, confirming p-LATS1/2^Thr1079/1041^ functional inhibition, and therefore de-repression of YAP/TAZ/TEAD association and transcriptional activity (Figure 5B). YAP/TAZ/TEAD target genes *CYR61*, *CTGF*, *AXL*, *TAZ* and *AREG*’s mRNA expression, which was repressed by LIF treatment, also tended to increase in the presence of XMU-MP-1. These LIF-induced effects were hindered when the inhibitor was added either in 2D (AGS cells) or 3D (MKN45 spheroids) culture conditions (Figure 5C). All this suggests that LIF-induced effects on gastric CSCs pass through Hippo core kinases activation and YAP/TAZ/TEAD effectors inhibition.

This was confirmed in functional tumoursphere-forming assays carried out in the presence of XMU-MP-1 and LIF, showing that when LATS1/2^Thr1079/1041^ phosphorylation is inhibited, LIF is no longer able to decrease the number of formed spheres and it even increases it in the AGS cell line (Figure 5D). This shows that in GC cell lines AGS and MKN45 as well as in PDX cells GC07, the LIF-induced anti-CSC effect involves the activation of the Hippo core kinases.

Moreover, as the LIF/LIFR canonical pathway is the JAK1/STAT3 pathway, the implication of this signalisation for the anti-CSC effects was evaluated. JAK1-induced STAT3^Tyr705^ phosphorylation was efficiently inhibited by the JAK1 inhibitor Ruxolitinib (Figure 5A(ii)) and no variation was noted in LATS2 and LATS1/2^Thr1079/1041^ phosphorylation (Appendix A). A significant decrease in the number of tumourspheres was observed in the presence of LIF and Ruxolitinib in the AGS, MKN45 and GC07 cells, compared with cells treated with Ruxolitinib only. JAK1/STAT3 inhibition did not affect LIF-induced anti-CSC effects in GC (Figure 5D), showing that this LIF canonical pathway is not involved in this phenomenon. Finally, double Hippo and JAK1/STAT3 inhibition resulted in the neutralisation of LIF-induced inhibition of tumoursphere formation, confirming the role of the Hippo pathway in LIF anti-CSC properties (Figure 5D) and suggesting an antagonistic role of the two pathways in response to LIF.

LIF-induced Hippo kinase activation and LIF anti-CSC effects thus seem to be linked. The LIF-induced decrease in CSC population and tumorigenic properties passes through the Hippo tumour suppressor MST1/2 and LATS1/2 kinases activation, with the consequent inhibition of Hippo YAP/TAZ/TEAD oncogenic effectors activity.

### 2.6. LIF-Induced Chemotherapy Potentiating Effects Pass through the Hippo Pathway

Since LIF was observed to potentiate conventional chemotherapy effects on gastric CSCs (Figure 3), the Hippo kinase inhibitor XMU-MP-1 was used to verifiy whether this effect of LIF also involved the Hippo pathway. Interestingly, LIF in combination with Doxorubicin was less efficient in decreasing tumoursphere-forming capacity when XMU-MP-1 was added to AGS, MKN45 as well as GC07 cells than in the absence of XMU-MP-1 (Figure 6A,B). The results are less obvious for the 5-FU treatment in AGS and MKN45 but remained similar in PDX cells GC07. Hippo kinase inhibition thus decreased LIF potentiating effects in chemotherapy, confirming this pathway’s involvement in LIF-anti-CSC effects. In addition, inhibition of the JAK1/STAT3 pathway did not affect LIF-induced reinforcement of chemotherapeutical effects in all cells (Figure 6A,B), showing that this pathway is not involved in this phenomenon.

LIF-induced chemotherapy potentiating effects thus also involve the Hippo pathway kinases and not its canonical pathway JAK1/STAT3.

### 2.7. LIFR Is Underexpressed in GC Tissue and Low LIF/LIFR is Associated to Poor Survival in Diffuse Type GC

The need for new reliable biomarkers in cancer is more and more important for proper diagnosis and to predict prognosis of patients, especially in GC, where relapse is high. Since LIF supplementation and activation of LIF/LIFR signalling in the GC context seems to be beneficial in terms of CSC targetting, its capacity as a GC cancer biomarker was pondered. Consequently, the Oncomine database was queried for LIFR expression in human GC tissues compared with associated non-tumorous gastric mucosa. Both the Cui and Cho datasets show that LIFR is significantly under-expressed in GC tissues versus associated non-tumorous tissues (Figure 7A(i)). Further analysis of LIFR according to the different GC subtypes showed that it is under-expressed in all subtypes without significant difference between diffuse, intestinal and mixed GC histological subtypes (Figure 7A(ii)).

Interestingly, analysis of GC patients’ overall survival probability using the KMplot database showed that in diffuse type GC, low LIF and LIFR expression are associated with low patient survival compared with high LIF and LIFR expression, which lead to a better prognosis of patients (Figure 7B). In intestinal type GC, this tendency remained the same for LIFR expression but is inversed for LIF since low LIF expression is correlated to high survival of patients and high expression to low survival (Figure 7B). This difference in prognosis of GC patients again shows how LIF is pleiotropic and can lead to different fates depending on the type of cancer. In addition, overall patients’ survival analysis according to LIFR and LIF combined expression showed that high expression of both in diffuse type GC leads to better prognosis than low expression, and the same trend is observed for intestinal type GC, though not significant.

Furthermore, patients’ median survival was analysed according to *YAP1*, *TAZ* and their target genes’ expression (Table 1). High expression of these Hippo tumorigenic genes is linked to low survival of GC patients. Interestingly, when the same analysis was carried out in GC patients having low LIFR expression, *YAP1*, *TAZ* and their target genes’ high expression leads to worse survival than low expression, and on the contrary, in high-LIFR-expressing patients, the less significant *p*-values show that this is not the case (Table 1). Finally, patients with higher expression of *YAP1*, *TAZ* and their target genes compared with LIFR expression have significantly lower survival rates than those with higher LIFR expression compared with Hippo oncogenic genes’ expression (Table 2). This interestingly confirms that LIF/LIFR signalling is of better prognosis in GC.

LIF/LIFR could thus represent a potential prognosis marker for diffuse type GC cases, a particularly aggressive type of GC.

## 3. Discussion

LIF/LIFR signalisation effect in cancer remains contradictory, with either negative or positive regulation of cancer cell properties depending on cancer types and studies [23,24,25,26]. In the GC context, LIF/LIFR signalling has been analysed only in few studies suggesting anti-tumorigenic effects [27,28], and none explored gastric CSCs.

This study focusses on the role of LIF/LIFR in CSCs, which are at the origin of GC initiation, progression and chemoresistance [4,5]. CSC targeting is of utmost importance in cancer therapy. Here, we show for the first time that LIF presents anti-CSC properties in both GC cell lines and PDX cells. LIF decreased one of the most important capacities of CSCs, which is the formation of small tumours in non-adherent conditions in vitro. It decreased the expression of the CD44 cell surface marker and ALDH activity, previously described as two of the best markers of gastric CSCs [5]. Moreover, consistent with previous studies allowing the analysis of different subpopulations of cells within GC cell lines and PDX cells [5], we hereby show that LIF acts more particularly on chemo-resistant CD44+ and ALDH+ Hoechst- CSCs by reducing their number and increasing the percentage of differentiated and non-tumorigenic CD44- and ALDH- Hoechst+ cells. Indeed, LIF has previously been described as an inducer of differentiation and inhibitor of proliferation of the myeloid leukemic cell line M1 [18] and cervical carcinoma cells [33,34]. In our AGS and MKN45 cells, LIF anti-CSC effects seems to be linked to an anti-proliferative effect (Appendix A) coherent with Xu et al.’s recent study showing that LIF inhibits proliferation of GC cells, among which MKN45 cells, through a G1-phase, arrest induction [28]. The anti-proliferation and pro-differentiation effects of LIF could thus explain the decrease in the population of CD44 + Hoescht+ and ALDH + Hoechst+ cells corresponding to CSCs and/or proliferating progenitor cells, as previously described [5], in favour of an increase in differentiated non-proliferating and non-CSC Hoechst+ cells observed in the present study. It should however not be left aside that the LIF anti-CSC effect could also be due to a decrease in CSC viability in the presence of LIF. Indeed, the cell viability assay using uptiblue reagent showed a decrease in viability of cells forming AGS spheres after LIF treatment (Appendix A).

Furthermore, we show that, in GC, LIF/LIFR signalling activates the Hippo kinase core MST1/2 and LATS1/2, thus inducing a phosphorylation cascade leading to the negative phosphorylation of oncogenic Hippo effectors YAP/TAZ and decreased oncogenic activity of the transcription factor TEAD, in accordance with what was observed in breast cancer [16,17].

Hippo effectors YAP/TAZ have been linked to CSC maintenance in several cancers including GC and, conversely, Hippo kinase activation is anti-tumorigenic [11,13,14,16]. Interestingly, Guo et al. have shown that LIF/LIFR pro-metastatic effects in melanoma pass through the activation of the canonical JAK/STAT pathway, which is known to be pro-oncogenic, and does not implicate the Hippo pathway. We hereby demonstrate that, in our model, LIF-induced anti-CSC effects involve Hippo pathway kinases activation since the inhibition of MST1/2-mediated LATS1/2^Thr1079/1041^ phosphorylation by XMU-MP-1 reverted LIF-induced anti-CSC effects. Hippo target genes’ downregulation by LIF in our 3D culture condition models confirms that Hippo effectors were inhibited in LIF-treated tumourspheres, which was counteracted in the presence of XMU-MP-1. Interestingly, some Hippo pathway regulators like RASSF1A are known for their pro-differentiation capacities. RASSF1A is an MST1/2 partner and activator with important roles in the promotion of differentiation phenotypes in cells, through the YAP-p73 transcriptional programme [35,36,37], and could be involved in the LIF/Hippo effects observed in this study, making its study an interesting perspective. While the JAK1/STAT3 pathway was concomitantly activated in response to LIF, experiments performed with JAK1 inhibitor Ruxolitinib showed that the JAK1/STAT3 pathway was not involved in the LIF-mediated anti-tumorigenic effect in GC.

LIF activation of the Hippo and JAK1/STAT3 pathways in GC seems to lead to antagonistic effects since LIF had no effect on CSC properties when double Hippo-JAK1 inhibitions were carried out. When added to GC cells, LIF seems to shift the activation balance of the JAK1/STAT3 and Hippo pathways towards a higher activation of Hippo kinases and a lower activation of JAK1/STAT3 signalisation. Azmal Ali et al. and Nandy et al. suggested connections between these two pathways with STAT3 regulation being downstream LIF/LIFR/Hippo [29,30]. They demonstrated that JAK/STAT3 pharmacological- and si-RNA-based inhibition did not change Hippo effectors expression levels, while inhibition of TAZ phosphorylation and thus higher nuclear translocation of TAZ led to induction of STAT3 expression and phosphorylation. This could be the same in our case with LIF/LIFR signalling activating Hippo kinases, thus inducing YAP/TAZ phosphorylation and reduced nuclear localisation, resulting in the possible under-regulation of both YAP/TAZ/TEAD and STAT3 oncogenic pathways. It should however be considered that LIF/LIFR solicits a large variety of signalisation pathways, which are more or less linked to the most studied JAK1/STAT3 pathway, and that signalisation cross-talks are very complex [22,38]. For instance, in HCC, LIF/LIFR anti-metastatic properties are due to the negative regulation of another LIF-regulated signalling pathway which is the PI3K/AKT pathway [24], again showing the complexity of LIF/LIFR signal transduction. LIF/LIFR/PI3K/AKT signalling has not been explored in the GC context.

LIF/LIFR signalling depends on the expression of LIFR GP190, which is an essential component of the heterodimeric LIF receptor GP130/GP190. Many studies have reported LIFR differential expression in the cancer context compared with non-tumoral conditions. Here, we show with in silico analyses that LIFR is under-expressed in GC tissues compared with adjacent non-tumorous tissues, suggesting a dysregulated LIF/LIFR signalisation in GC. Further, there was no difference in LIFR expression between the different histological subtypes according to the Lauren classification of GC, but when overall survival of patients was compared, diffuse type GC patients with low LIF and LIFR expressions had worser prognosis than those with higher LIF and LIFR expressions. Consistent with this, Zhao et al. described a low expression of long non-coding RNA-LOWEG in GC which normally upregulates LIFR [27]. This LIFR low expression could be a potential prognostic marker in GC, especially for the diffuse type cases where it reflects a low prognosis. Moreover, Zhao et al. also described LOWEG as having an anti-invasion capacity in GC and suggested a possible link between these anti-invasive effects and LOWEG-induced LIFR upregulation capacity [27]. Many other studies describing LIF/LIFR as anti-tumorigenic have also suggested an anti-metastatic role [16,24,31]. An interesting perspective could thus be to further study the role of LIF on the invasive properties of CSC in GC and most particularly on their metastatic properties.

This study presents LIF as a potential anti-CSC therapy in GC. Interestingly, we found that LIF was downregulated in CD44+ PDX cells corresponding to CSCs compared with non-CSC CD44- cells. Interestingly, Xu et al. showed that the LIF protein expression level is downregulated in GC, and more importantly, that moderately and well-differentiated intestinal type GC had more LIF-positive cells compared with poorly differentiated diffuse-type GC. In addition, LIF/LIFR combined high expression in diffuse-type GC is correlated to better patients’ prognosis compared with low expression. This comforts the hypothesis of LIF as a possible treatment in GC, to target chemo-resistant CSCs and palliate LIF low expression in diffuse GC cases, which are of poor prognosis. Interestingly, here we show that LIF is able to potentiate the effect of chemotherapeutical drugs on gastric CSCs of both GC cell lines and PDX cells, and that LIF-induced chemotherapy potentiating effects pass through Hippo kinase activation by LIF. CSCs are well-known for their capacity to resist conventional chemo/radiotherapies and form the small pool of residual cells responsible for post-treatment tumour relapse. LIF could be used to improve cancer therapies via its anti-CSC and chemo-potentiator properties, thus reinforcing the hypothesis of using it in GC treatment.

The use of LIF in vivo is however questionable due to its highly pleiotropic aspect. Several in vivo attempts can be found in the literature in which LIF injection or overexpression via implantation of LIF-producing cells in mouse models can promote anti-inflammatory signalling through IL-12 [39], suppress type 2 immunity in muscular dystrophic mouse model [40] and inhibit Th17 response in inflamed colon of inflammatory bowel disease models [41]. In addition, the commercial human LIF Emfilermin^TM^ was used in a phase II clinical trial for the prevention of chemotherapy-induced peripheral neuropathy [42]. Despite the lack of positive response to LIF, patients showed no side effects, confirming the promising use of LIF for systemic treatments. Moreover, a recent study designed and synthetised LIF-loaded and vectorised nanoparticles for the targeting of activated peripheral macrophages, which were able to decrease murine leukemic cell M1 proliferation [43].

This strategy could be adapted to target gastric CSCs more specifically. Finally, the use of PDX cells and the anti-CSC effect of LIF in this model further contributes to the translational potential of this study.

## 4. Materials and Methods

### 4.1. Cancer Cell Lines and Patient-Derived Xenograft (PDX) Cell Culture

AGS and MKN45 diffuse GC cell lines were cultured in DMEM F12-Glutamax and RPMI 1640-Glutamax media, respectively, supplemented with 10% heat-inactivated foetal bovine serum (FBS) (all from ThermoFisher Scientific, Villebon sur Yvette, France) and 50 µg/mL vancomycin (Invitrogen, Cergy-Pontoise, France) at 37 °C in a 5% CO_2_ humidified atmosphere. All cell lines, authenticated by STR profiling, were mycoplasma-free after PCR verification. AGS cells (ATCC CRL-1739^TM^) were derived from primary gastric carcinoma. Mutations found in this cell line concern CDH1 (c.1733_1734insC), CTNNB1 (c.101G > A), KRAS (c.35G > A) and PIK3CA (c.1357G > A). MKN45 is a poorly differentiated gastric adenocarcinoma cell line, derived from liver metastases. It is mutated for CDH1 (c.823_832 + 8del), BRCA1 (c.4792T > C), ERBB4 (c.82 + 8C > T) and TP53 (c.328C > T). Patient-derived xenograft (PDX) cells GC04, GC06, GC07 and GC10 were established through serial subcutaneous xenografts in NSG mice [5]. Animals were maintained at the University of Bordeaux Animal facility in accordance with institutional animal use and care committee guidelines (accreditation number B33-063-916, received on 23 May 2016). All animal experimentations were approved by the French Ethics Committee on Animal Experiments CEEA50 of Bordeaux (Authorisation n° A12005, ref 2017103118319700 v7). Molecular analysis was carried out, according to standard methods on PDX tumour tissue sections (in situ hybridisation) and on extracted DNA to determine Epstein–Barr virus (EBV) status. GC04, GC06, GC07 and GC10 were all EBV-negative. NSG colon and lung panels were used to search for mutations for KRAS, EGFR, BRAF, PI3KCA, AKT1, ERBB2, PTEN, NRAS, STK11, MAP2K1, ALK, DDR2, CTNNB1, MET, TP53, SMAD4, FBX7, FGFR3, NOTCH1, ERBB4, FGFR1 and FGFR2. GC04 is mutated for PIK3CA (c.1633G > A p. (Glu545Lys)), GC06 for KRAS (c.175G > A p. (Ala59Thr) and c.38G > A p. (Gly13Asp)), GC07 for STK11 (c.1019A > G p. (Tyr340Cys)) and GC10 for TP53 (c.818G > A p. (Arg273His)).

When the tumour sizes reached 200–400 mm^3^, they were recovered and dissociated using a human tumour dissociation kit in C tubes and GentleMACS dissociator following manufacturer recommendations (All from MACS, Miltenyi, Paris, France). Cells were successively filtered using 70 and 40 µm cell strainers and red blood cells were lysed. Cells could then be processed for tumoursphere formation assays and RNA experiments.

### 4.2. Tumoursphere Culture

AGS (500 cells per well), MKN45 (100 cells per well) and PDX (1000 cells per well) cells were seeded in 96-well culture plates, previously coated with a 10% poly-2-hydroxyethyl methacrylate (Sigma-Aldrich, Saint-Quentin Fallavier, France) solution in 95% (*v/v*) ethanol and left to dry overnight at 50–60 °C to make them non-adherent. Cells were incubated at 37 °C in a 5% CO_2_, humidified atmosphere in DMEM F12-Glutamax medium, supplemented with 0.3% glucose, 1:100 N2-supplement (all from ThermoFisher Scientific, Villebon sur Yvette, France), 20 ng/mL human epithelial growth factor, 20 ng/mL basic fibroblast growth factor and 5 μg/mL insulin (all from Sigma-Aldrich, Saint-Quentin Fallavier, France). MKN45 tumourspheres were grown in serum-free medium while AGS and PDX cells needed 2% FBS supplementation to grow.

### 4.3. Leukaemia Inhibitory Factor, Hippo Inhibitor and JAK/STAT Inhibitor Treatments

Recombinant human leukaemia inhibitory factor (LIF), produced in *E. coli*, was purchased from PeproTech (Neuilly-Sur-Seine, France), reconstituted in PBS and stored at −80 °C. It was used at 50 ng/mL in most experiments and 10 ng/mL in some tumoursphere assays. Adherent cells were treated after medium change for serum starvation. LIF was added either in a time-course manner (0.5, 2, 5, 24 and 48 h) or only at particular times depending on the experiments. Hippo inhibitor XMU-MP-1 (Selleckchem, Euromedex, Souffelweyersheim, France) was diluted in DMSO, stored at −80 °C and used at 0.5 µM. Ruxolitinib (Stemcell Technologies, Grenoble, France) JAK1/STAT3 inhibitor was also diluted in DMSO and stored at −20 °C before use at 1 µM. Treatments with inhibitors were carried out 30 min before LIF treatments.

### 4.4. Tumoursphere-Forming Assays and Chemotherapy Tests

AGS, GC06, GC10 and GC04 cells were seeded in non-adherent 3D conditions as described above and were immediately treated with 0, 10 and/or 50 ng/mL LIF or XMU-MP-1 and Ruxolitinib before LIF. Two hundred thousand MKN45 and GC07 cells were first plated in adherent conditions in 6-well plates and serum-deprived the day after before treating with LIF for 48 h. Cells were then trypsinised before plating in 3D conditions as described above. All cells’ LIF and/or inhibitor treatments were repeated every 48 h before counting the number of spheroids per well at 7–8 days after the first treatment under an inverted light microscope using a ×20 objective.

For chemotherapy tests, chemotherapy drugs Doxorubicin 0.1 µM and 5-Fluorouracil 50 µM (both from Sigma-Aldrich, Saint-Quentin Fallavier, France), in combination or not with LIF, were added to AGS, MKN45 and GC07 cells upon seeding in non-adherent conditions and formed spheroids were recorded as mentioned above.

### 4.5. Flow Cytometry

AGS, MKN45 and GC07 cells were plated (100,000 cells) in 12-well plates, serum-deprived the day after and treated or not with 50 ng/mL LIF for 48 h. Cells were incubated with 1:25 mouse anti-human CD44-APC antibody (G44-26 clone, BD Biosciences, Le Pont de Claix, France) or isotype controls in ice-cold buffer containing PBS-0.5% bovine serum albumin (BSA, Gibco, ThermoFisher Scientific, Villebon sur Yvette, France)-2 mM EDTA (Sigma-Aldrich, Saint-Quentin Fallavier, France) for 25 min at 4 °C. ALDEFLUOR Kit (STEMCELL Technologies, Grenoble, France) was used in MKN45 cells prior to CD44 staining to detect ALDH activity, according to the manufacturer’s instructions. Cells were rinsed thrice with ice-cold buffer and incubated for 10 min with buffer containing 50 µg/mL 7-Aminoactinomycin-D (7-AAD 559925, BD Biosciences, Le Pont de Claix, France). LIFR cytometry preparation was carried out as above with anti-human CD118 (LIFR)-PE antibody (12D3 clone, BD Biosciences, Le Pont de Claix, France). Cytometry analysis was performed using a BD FACSCanto II instrument and the DIVA software (BD Biosciences, Le Pont de Claix, France). Dead cells were excluded based on side scatter characteristics and 7-AAD positivity.

### 4.6. Luciferase Reporter Assay

AGS and MKN45 cell lines were seeded (50,000 cells) in 24-well plates and transfected with 100 ng/well TEAD (8XGTII-luciferase) or TATA box control (transcription-activator like TAL) firefly luciferase reporters (BD Biosciences, Le Pont de Claix, France) along with 10 ng/mL renilla luciferase reporter pRL-SV40 (Promega, Charbonnieres Les Bains, France). Serum starvation was then carried out before LIF and/or inhibitor treatment at different time intervals (0.5, 2, 5, 25 and/or 48 h). Firefly and renilla luciferases activities were measured using a Dual Luciferase assay kit (Promega, Charbonnieres Les Bains, France) and firefly luciferase activity values were normalised by the renilla luciferase activity values for each sample for transfection efficiency. TEAD-specific luciferase activity was then normalised by that of TAL.

### 4.7. Immunofluorescence Assay

AGS and MKN45 cell lines were seeded (50,000 cells) on glass coverslips in 24-well plates, serum-deprived 24 h later and treated with LIF at different time intervals (0.5, 2, 5, 24 and/or 48 h). Cells were then fixed with 3% paraformaldehyde solution in cytoskeletal buffer prior to immunofluorescence staining. After permeabilisation with 0.1% Triton X-100 solution for 1 min and blocking in 1%-BSA 2%-FBS in tris-buffered saline (TBS) solution, cells were stepwise incubated in corresponding primary antibodies for 1 h, washed 3 times in TBS and left for 30 min in secondary antibody coupled with 1:350 Alexa Fluor^®^ 488 goat anti-rabbit or anti-mouse IgG antibodies, 1:250 Alexa Fluor^®^ 647-coupled Phalloidin and 50 mg/mL 4′-6-diamino-phenyl-indol (DAPI) (all from ThermoFisher Scientific, Villebon sur Yvette, France). Primary antibodies used were 1:100 rabbit anti-YAP and 1:100 mouse anti-TAZ E5P2N (cat.4912 and cat.71192, respectively; Cell Signalling Technology, Saint-Cyr-L’École, France). Images were taken using an Eclipse 50i epi-fluorescence microscope (Nikon, Champigny sur Marne, France) with the NIS-BR acquisition software and a ×40 (numerical aperture, 1.3) oil immersion objective. Nuclear YAP/TAZ count and mean grey value measurements were carried out using the ImageJ 1.52p software (National Institutes of Health, Rockville Pike, Bethesda, MA, USA) [44].

### 4.8. Live Immunofluorescence Assay

MKN45 (5000) cells were plated in non-adherent 6-well plates and 5-day-old spheres were treated or not for 48 h with LIF before harvesting. ALDEFLUOR Kit (STEMCELL Technologies, Grenoble, France) was used to detect ALDH activity, according to the manufacturer’s instructions, prior to spheres incubation with 1:25 anti-human CD44-PE antibody (515 clone, BD Biosciences, Le Pont de Claix, France) in ice-cold buffer containing PBS-0.5% bovine serum albumin (BSA, Gibco, ThermoFisher Scientific, Villebon sur Yvette, France)-2 mM EDTA (Sigma-Aldrich, Saint-Quentin Fallavier, France) for 25 min at 4 °C. Spheres were rinsed twice with ice-cold buffer and incubated for 30 min at RT with Hoescht-33342 dye (ThermoFisher Scientific, Villebon sur Yvette, France). Live immunofluorescence acquisition was carried out with an Eclipse 50i epi-fluorescence microscope (Nikon, Champigny sur Marne, France) using the NIS-BR acquisition software and a ×40 objective (numerical aperture, 1.3) immediately after buffer-washing and slide mounting. CSC subcellular populations were analysed using the ImageJ 1.52p software (National Institutes of Health) [44].

### 4.9. RNA Extraction and RTqPCR

For the 2D-RNA analysis, cells were plated (100,000 cells) in 12-well plates and serum-deprived the day after, and for 3D-RNA experiments, 2000 cells were seeded in non-adherent conditions for 4–5 days prior to treatments. All cells were treated or not with 50 ng/mL LIF either at different time intervals (0.5, 2, 5, 24 and 48 h) or at particular times. Total RNA was extracted using TRIzol^TM^ reagent (ThermoFisher Scientific, Villebon sur Yvette, France) according to manufacturer’s recommendations. Reverse transcription was then carried out with the Quantitech Reverse Transcription kit (QIagen, Germantown, Netherlands) following the manufacturer’s protocol. Real-time PCR was performed using SYBR-qPCR-Premix Ex-Taq (Takara, Shiga, Japan) and 0.5 µM specific primers (Appendix A). Analysis of amplified RNA samples was carried out using the 2^−∆∆CT^ method with TBP and HPRT1 as normalisers.

### 4.10. Agilent Microarray

PDX cells (*n* = 4, in simplicates and *n* = 2, in triplicates) were FACS-sorted according to CD44 expression. DNA was removed and RNAs extraction was carried out using an RNeasy microkit (all from Qiagen, Germantown, the Netherlands). RNA quantification was carried out using the TapeStation system (Agilent) and RNA integrity numbers (RIN) were determined. Gene expression profiling was performed at the GeT-TRiX facility (GenoToul, Génopole Toulouse Midi-Pyrénées, Toulouse, France) with Agilent Sureprint G3 Human microarrays (8 × 60K, design 072363), following the manufacturer’s instructions. One-Color Quick Amp Labelling kit (Agilent) and Agencourt RNAClean XP (Agencourt Bioscience Corporation, Beverly, MA, USA) were used to prepare cyanine-3 (Cy3)-labelled cRNA from 25 ng of total RNA of each sample, according to the manufacturer’s conditions. Dye incorporation and cRNA yield were then verified using a Dropsense™ 96 UV/VIS droplet reader (Trinean, Ghent, Belgium). Cy3-labelled cRNA (600 ng) were hybridised on microarray slides following the manufacturer’s instructions. Slides were washed and scanned immediately using an Agilent G2505C Microarray Scanner and the Agilent Scan Control A.8.5.1 software. The Agilent Feature Extraction software v10.10.1.1 was used with default parameters for fluorescence signal extraction and only the most expressed probes values were represented. T-test statistical analysis was carried out to compare differentially expressed genes.

### 4.11. Protein Extraction and Western Blotting

Cells were plated (200,000 cells) in 6-well plates, serum-deprived the day after and treated or not with 50 ng/mL LIF either at different time intervals (0.5, 2, 5, 24 and 48 h) or at specific times. Cells were then washed twice with ice-cold PBS and lysed with ProteoJet reagent (ThermoFisher Scientific, Villebon sur Yvette, France) supplemented with a cocktail of protease inhibitor P8340, and phosphatase inhibitors P0044 and P5726 (all from Sigma-Aldrich, Saint-Quentin Fallavier, France). Total proteins were submitted to 7.5% or 12% sodium dodecyl sulphate polyacrylamide gel electrophoresis (SDS-PAGE) and transferred onto nitrocellulose membranes. Membranes were blocked in 5% BSA-0.1%PBS-Tween 20 and incubated with respective primary antibodies for 1 h at room temperature (RT). Primary antibodies used at 1:2000 were: rabbit anti-YAP, anti-p-YAP^Ser127^, mouse anti-TAZ E5P2N, anti–LATS2 D83D6, anti-p-LATS1/2^Thr1079/1041^ D57D3, mouse anti-STAT3 124H6, p-STAT3^Tyr705^ D3A7 (cat.4912, cat.4911, cat.71192, cat.5888, cat.8654, cat.9139 and cat.9145, respectively; Cell Signaling Technology, Saint-Cyr-L’École, France), mouse anti–α-tubulin (cat.T-6074; Sigma-Aldrich, Saint-Quentin Fallavier, France) and mouse anti–glyceraldehyde-3-phosphate dehydrogenase (cat.sc-47724; Santa Cruz Biotechnology, Heidelberg, Germany). Membranes were then probed with starbright Blue700 goat anti-mouse and anti-rabbit fluorescence conjugated secondary antibodies (Bio-Rad, Marnes-la-Coquette, France) and diluted at 1:2500 for 1 h at RT. Antibody-bound proteins were detected using ChemiDoc™ Imaging Systems (Bio-Rad, Marnes-la-Coquette, France). Immunoblotting band intensity was measured using the ImageJ 1.52p software (National Institutes of Health, USA) [44].

### 4.12. In Silico Database Analysis and Statistics

*LIFR* mRNA expression patterns from GC were extracted from the Oncomine database tool (www.oncomine.org) [45]. Results were filtered for cancer versus normal tissue analysis and/or Lauren classification and gastric cancer cases in the primary filter. Cui (*n* = 90) and Cho (*n* = 160) Gastric tissue datasets were chosen and extracted for analysis.

The KMplot database tool (www.kmplot.com) [46] was used to analyse overall survival probability of non-metastatic (parameter M = 0) GC patients (*n* = 444) according to LIFR, LIF and combined LIFR and LIF expression. The following JetSet best probes, 225575_at (LIFR) and 205266_at (LIF), were used. Analysis was filtered for Lauren classification, to analyse intestinal and diffuse types of GC separately, and the GSE62254 dataset was excluded as suggested by the database to avoid biased results. Patients’ samples were separated according to high or low expressions by the software’s best cut-off value auto-setting. *P*-values were calculated by a log-rank test.

Results are expressed as mean ± S.E.M of at least three independent experiments. Statistical tests were carried out using the GraphPad Prism software version 8.0.2 (La Jolla, CA, USA). Mann–Whitney test or Student *t*-test were used for the two-groups comparisons and ANOVA with Bonferroni as a post hoc test or Kruskal–Wallis test with Dunn’s test as post hoc were performed for multiple comparisons.

## 5. Conclusions

This study deciphers for the first time the role of LIF in the GC context and more specifically on the signalling pathways controlling gastric CSCs’ tumorigenic properties. We show that LIF is anti-tumorigenic in GC cell lines and most importantly in patient-derived GC cells, opening further translational perspectives. Signalling pathways and molecular mechanisms involved were addressed, and the Hippo pathway was found to be at the basis of LIF-induced anti-CSC effects in GC. In this regard, this work reveals LIF as an interesting anti-CSC therapeutic potential in combination with conventional chemotherapy, which is of utmost important in this bad prognosis disease. LIF/LIFR signalling hereby represents a possible prognosis marker and therapeutic option in the management of GC.

## Figures and Tables

**Figure 1 cancers-12-02011-f001:**
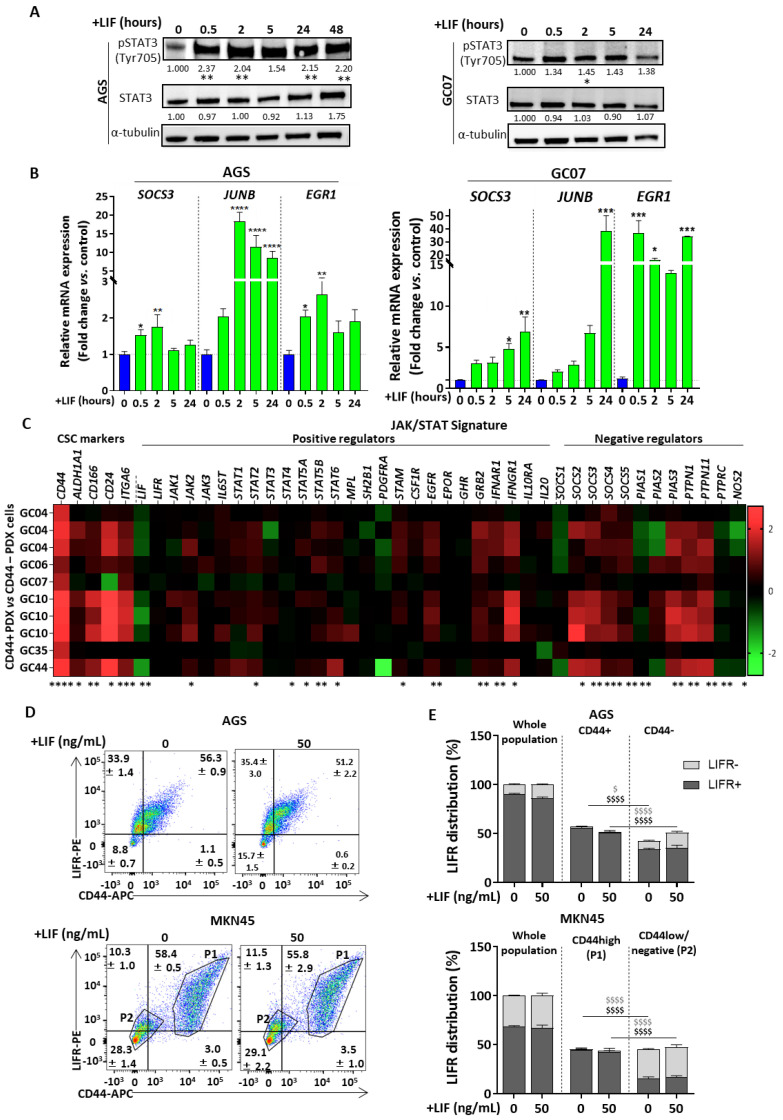
The Leukaemia inhibitory factor (LIF)/LIF receptor (LIFR)/JAK/STAT pathway is functional in gastric cancer (GC) and upregulated in gastric cancer stem cells (CSCs). (**A**) p-STAT3^Tyr705^ and STAT3 protein levels in AGS and MKN45 cell lines and GC07 patient-derived xenograft (PDX) cells after time course treatment with 50 ng/mL LIF. Values under each band represent quantification of relative tubulin-normalised protein expression according to band density (whole Western Blot available in Appendix A) (**B**) JAK/STAT targets relative mRNA expressions after treatment of AGS, MKN45 and GC07 cells with (green) or without (blue) LIF. (**C**) Relative FACS-sorted CD44+ PDX cells versus CD44- PDX cells gene expression profiles. Three groups are represented: CSC markers, JAK/STAT positive and negative regulators. (**D**) Flow cytometry analysis of LIFR-GP190 protein expression in CD44+/high cells and CD44-/low cells after a 48 h LIF treatment of AGS and MKN45 cells. Mean +/− SEM is represented in each quadrant of the dot plot graphs. For MKN45 cells, P1 correspond to CD44+/high population and P2 to CD44-/low population. (**E**) LIFR distribution in whole cell population, CD44+/high and CD44-/low cell subpopulations obtained from cytometry analysis in Figure 1D. LIFR+ cells are represented by dark grey bars and LIFR- cells by stacked light grey bars. The cumulation of both types of bars represents either the whole population, CD44+/high or CD44- cells. LIF treatments (50 ng/mL) were carried out at either different time intervals, 0, 0.5, 2, 5, 24 h (**A**,**B**) or for 48 h (**A**,**B**,**D**,**E**), 3 < *n* < 4. * *p* < 0.05, ** *p* < 0.005, *** *p* < 0.0005 and **** *p* < 0.0001 versus untreated controls with ANOVA statistical analyses. $ *p* < 0.05, $$ *p* < 0.005, $$$ *p* < 0.0005 and $$$$ *p* < 0.0001 versus corresponding CD44+/high cells with 2-way ANOVA tests. LIFR+ stats are represented by dark grey $ and LIFR- cells by light grey $.

**Figure 2 cancers-12-02011-f002:**
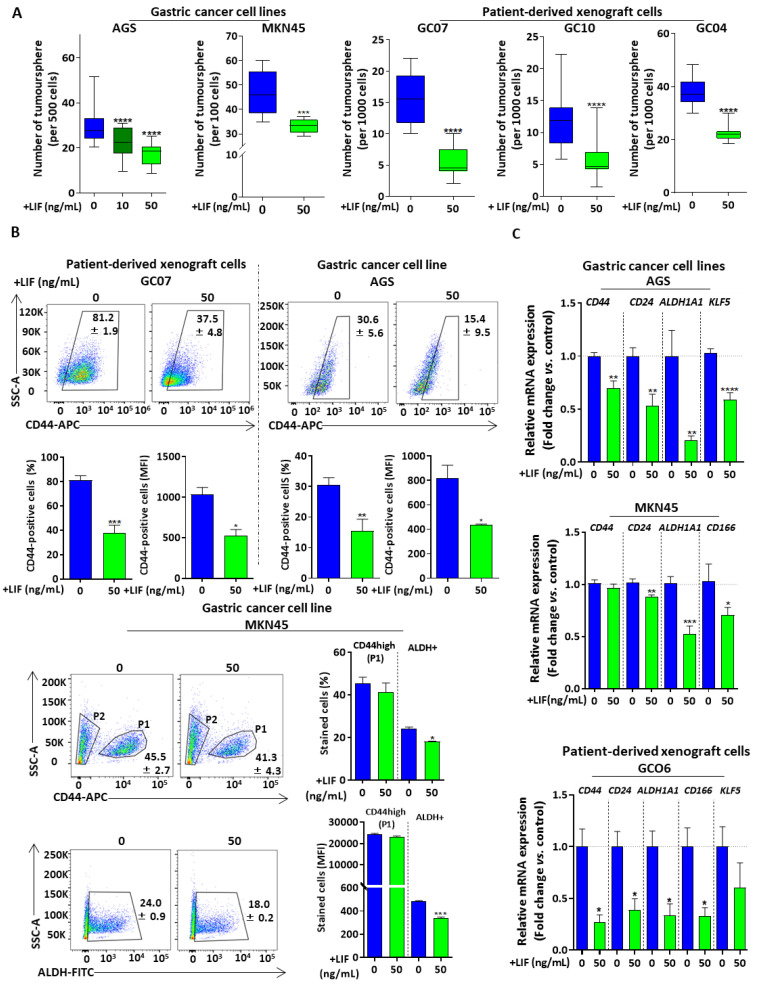
Leukaemia inhibitory factor presents anti-CSC effects in GC. (**A**) 3D tumoursphere assays carried out on GC cell lines (AGS and MKN45, on the left side of the panel) and PDX cells (GC07, GC10 and GC04 on the right side of the panel). (**B**) Dot plot representation (upper panel) and quantification (lower panel) of flow cytometry analysis of gastric CSC markers CD44 and ALDH’s activity. Mean +/− SEM is presented in each quadrant of the dot plot graphs. For MKN45 cells, P1 corresponds to CD44+/high population and P2 to CD44-/low population. (**C**) Relative mRNA levels of gastric CSC markers of PDX cells GC06 and GC cell lines AGS and MKN45. All cells were treated (green) or not (blue) with 50 ng/mL LIF for 48 h. For tumoursphere assays, LIF treatment was carried out every 48 h and sphere counting performed after 7 days. 3 < *n* < 5, * *p* < 0.05, ** *p* < 0.005, *** *p* < 0.0005 and **** *p* < 0.0001 versus. untreated controls with Mann–Whitney and Student *t*-tests.

**Figure 3 cancers-12-02011-f003:**
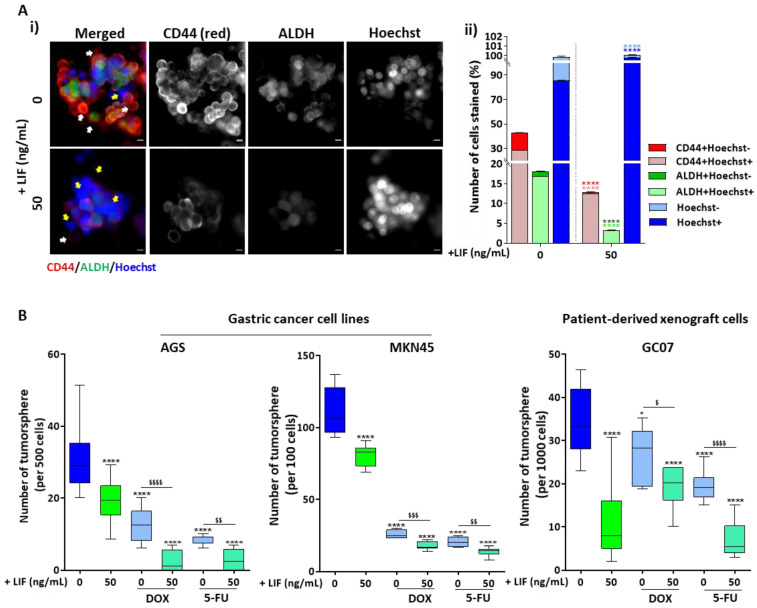
LIF/LIFR signalling potentiates chemotherapy effect on gastric CSCs. (**A**) Representative immunofluorescence images (**i**) and quantification (**ii**) of live MKN45 7-day-old spheres stained with CD44, Aldefluor reagent and Hoechst-33342 compound. White arrows point towards chemo-resistant CD44+/ALDH+Hoechst- cells and yellow arrows point towards differentiated non-CSC CD44-/ALDH-Hoechst+ cells. (**B**) 3D tumoursphere assays carried out on gastric cancer cell lines and patient-derived xenograft cells, treated (green) or not (blue) with 50 ng/mL LIF for 48 h. Cells were also treated or not with chemotherapy drugs Doxorubicin (DOX) and 5-Fluorouracil (5-FU). Cells treated with chemotherapy only are represented in light blue and those with combined LIF and chemotherapy treatments are in light green. LIF treatments were carried out every 48 h and tumoursphere counting was performed after 7 days. *n* = 3, * *p* < 0.05, ** *p* < 0.005, *** *p* < 0.0005 and **** *p* < 0.0001 versus respective untreated controls with ANOVA statistical analyses, colours representing the matched coloured bars comparison (**A**(**ii**)). $ *p* < 0.05, $$ *p* < 0.005, $$$ *p* < 0.0005 and $$$$ *p* < 0.0001 versus conditions treated with DOX or 5-FU alone with Student *t*-tests.

**Figure 4 cancers-12-02011-f004:**
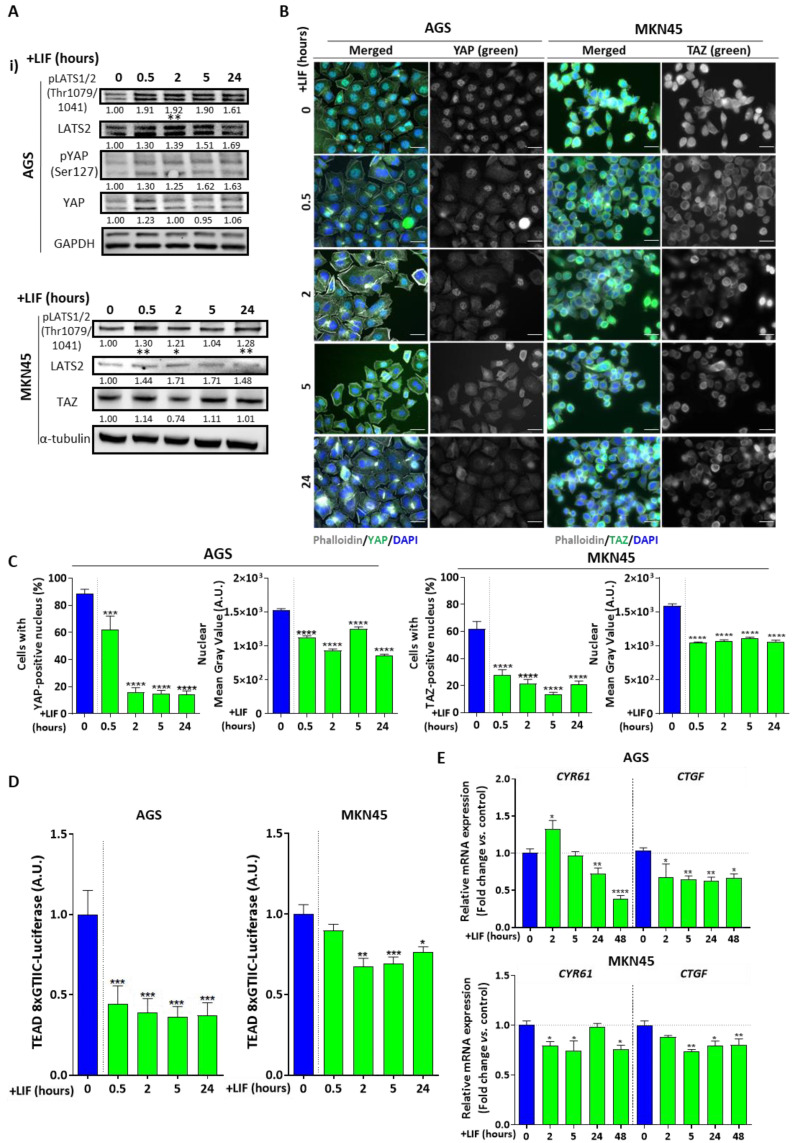
Leukaemia inhibitory factor activates the Hippo pathway tumour suppressor core. (**A**) Hippo suppressors LATS2 and p-LATS1/2^Thr1079/1041^ and Hippo effectors YAP, p-YAP^Ser127^ and TAZ’s protein expression in AGS and MKN45 cell lines. Values under each band represent quantification of relative GAPDH or tubulin-normalised protein expression according to band density (whole Western Blot available in Appendix A). (**B**) Representative immunofluorescence images of AGS and MKN45 cells stained with anti-YAP or anti-TAZ antibodies (green). All cells were marked with phalloidin (grey) and DAPI (blue). Scale bars 10 µm. (**C**) Relative quantification of cells with YAP- or TAZ-positive nucleus and respective mean grey values. (**D**) TEAD 8xCTIIC-luciferase reporter assay showing activity of transcription factor TEAD in AGS and MKN45 cell lines. (**E**) Relative Hippo target genes mRNA levels in AGS (2D) and MKN45 (3D) cell lines. All cells were either untreated (blue) or treated with 50 ng/mL LIF (green) at different time intervals (0, 0.5, 2, 5, 24, 48 h). * *p* < 0.05, ** *p* < 0.005, *** *p* < 0.0005 and **** *p* < 0.0001 versus untreated controls with ANOVA statistical analyses.

**Figure 5 cancers-12-02011-f005:**
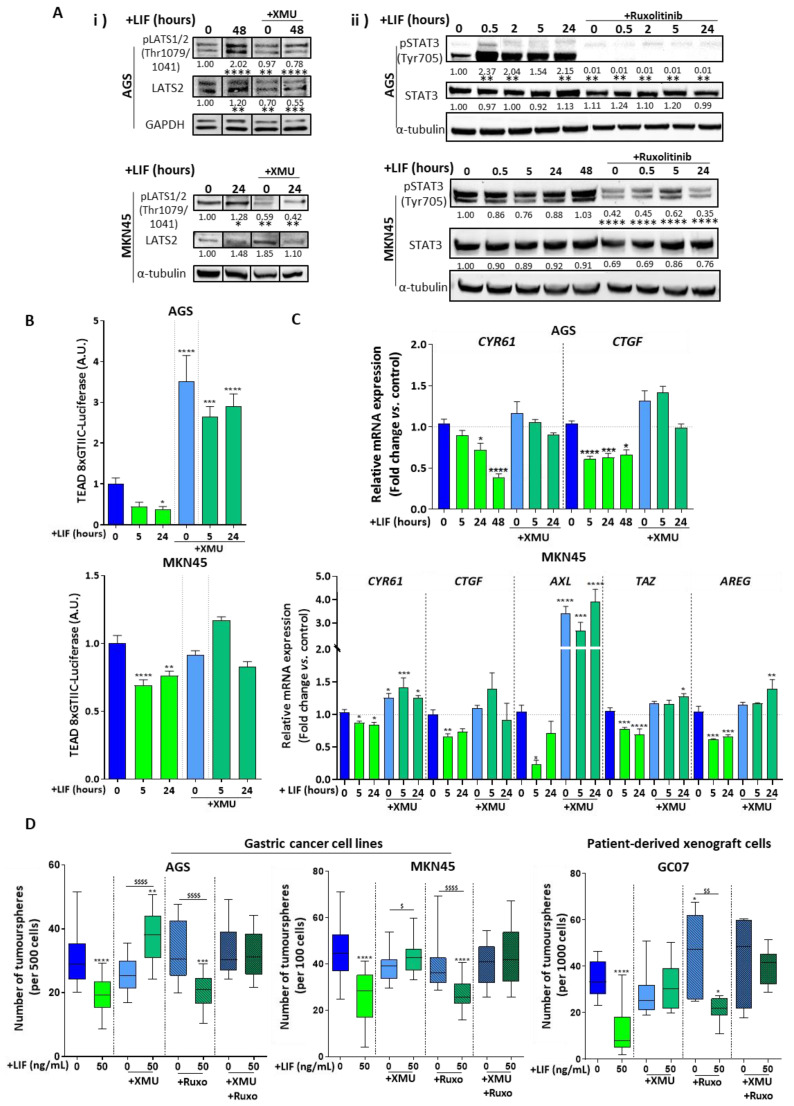
LIF anti-CSC effects are linked to Hippo pathway core kinases activation. (**A**) Hippo kinase LATS2 and p-LATS1/2^Thr1079/1041^ and JAK/STAT effector STAT3 and p-STAT3^Tyr705^ protein expression in AGS and MKN45 cell lines. Values under each band represent quantification of relative GAPDH or tubulin-normalised protein expression according to band density (whole Western Blot available in Appendix A). (**B**) TEAD 8xCTIIC-luciferase reporter assay showing activity of transcription factor TEAD in AGS and MKN45 cell lines. (**C**) Relative Hippo target genes mRNA levels in AGS (2D) and MKN45 (3D) cell lines. (**D**) 3D tumoursphere assays carried out on GC cell lines (AGS and MKN45, on the left side of the panel) and PDX cells (GC07, on the right side of the panel). All cells were treated with 50 ng/mL LIF (green) and/or 0.5 µM XMU-MP-1 (XMU) (emerald green) at different time intervals (0, 2, 5, 24 or 48 h). For tumoursphere assays, LIF treatment was carried out every 48 h, 1 µM JAK1 inhibitor Ruxolitinib (Ruxo) (hatched bars) and combination of both XMU and Ruxo (checked bars) was used and sphere counting was performed after 7 days. For each experiment, inhibitors were added 30 min before each LIF stimulation. * *p* < 0.05, ** *p* < 0.005, *** *p* < 0.0005 and **** *p* < 0.0001 versus untreated controls with ANOVA statistical analyses. $ *p* < 0.05, $$ *p* < 0.005, $$$ *p* < 0.0005 and $$$$ *p* < 0.0001 versus conditions treated with XMU-MP-1 and/or Ruxolitinib alone with Student *t*-tests.

**Figure 6 cancers-12-02011-f006:**
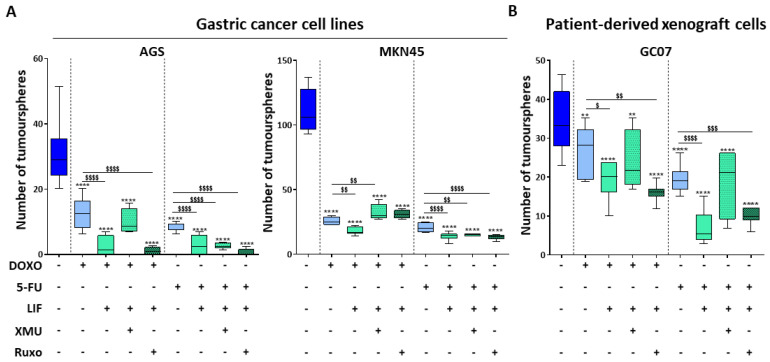
LIF-induced chemotherapy potentiating anti-CSC effects pass through the Hippo pathway. (A-B) 3D tumoursphere assays carried out on GC cell lines (**A**) and patient-derived xenograft cells (**B**), treated (green) or not (blue) with 50 ng/mL LIF for 48 h. Cells were also treated or not with chemotherapy drugs Doxorubicin (DOX) and 5-Fluorouracil (5-FU), Hippo kinase inhibitor XMU-MP-1 (XMU) and JAK1 inhibitor Ruxolitinib (Ruxo). Cells treated with chemotherapy drugs only are represented in light blue, while cells treated with LIF in combination with chemotherapy drugs and/or inhibitors are in light green. XMU-treated conditions are represented by dotted bars and Ruxo-treated conditions by checked bars. Treatments were carried out every 48 h and tumoursphere counting was performed after 7 days. * *p* < 0.05, ** *p* < 0.005, *** *p* < 0.0005 and **** *p* < 0.0001 versus untreated controls with ANOVA statistical analyses. $ *p* < 0.05, $$ *p* < 0.005, $$$ *p* < 0.0005 and $$$$ *p* < 0.0001 versus conditions treated with DOX or 5-FU alone with Student *t*-tests.

**Figure 7 cancers-12-02011-f007:**
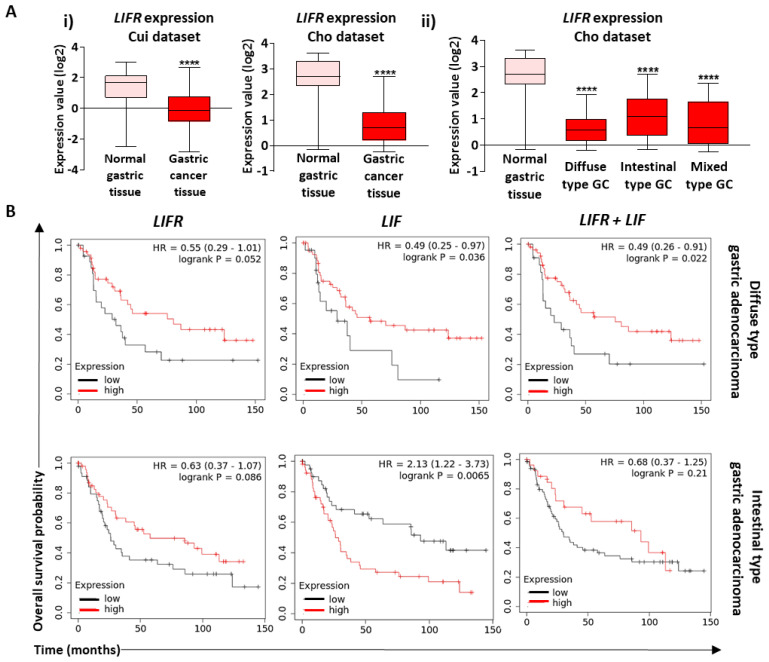
LIF and LIFR in GC and associated prognosis. (**A**) Oncomine data-mining analysis showing level of LIFR mRNA in (**i**) gastric adenocarcinoma compared with normal gastric mucosa in the Cui (*n* = 90) and Cho (*n* = 160) gastric tissue datasets and (**ii**) normal tissue compared with gastric cancer Lauren classification subsets (diffuse, intestinal and mixed type GC) in the Cho dataset. (**B**) KMplot database analysis showing overall survival probability of diffuse type and intestinal type gastric adenocarcinoma patients (23 < *n* < 175) according to LIFR and LIF and combined LIFR and LIF (LIFR + LIF) expression levels. * *p* < 0.05, ** *p* < 0.005, *** *p* < 0.0005 and **** *p* < 0.0001 versus untreated controls with Student *t*-tests and log-rank test.

**Table 1 cancers-12-02011-t001:** Patients’ median survival according to expression of YAP1, TAZ and their target genes according to LIFR expression or not.

Gene	Patients’ Median Survival (Months)								
	All Patients Independent of LIFR Expression			Patients with Low LIFR Expression			Patients with High LIFR Expression		
	High Expression	Low Expression	*p*-Value	High Expression	Low Expression	*p*-Value	High Expression	Low Expression	*p*-Value
*YAP1*	35.4	99.4	0.0011 **	21.4	41.2	0.00032 ***	36.4	113.2	0.013 **
*TAZ*	35.9	46	0.085	26.2	76.2	0.009 **	32.6	87	0.24
*CTGF*	35.9	85.6	0.017 *	15.2	30	0.022 *	32.6	93.2	0.0039 **
*CYR61*	35.9	45.8	0.39	45.1	32.1	0.36	44.7	87	0.12
*AXL*	36.4	85.6	0.04 *	26	76.2	0.027 *	36.4	99.4	0.083
*AREG*	36.4	78.6	0.1	32.1	40	0.27	75.5	25.2	0.11

**p* < 0.05, ***p* < 0.005, ****p* < 0.0005 vs. untreated controls with Log-Rank test.

**Table 2 cancers-12-02011-t002:** Patients’ median survival according to whether expression of YAP1, TAZ and their target genes is higher or lower than that of LIFR.

Gene	Patients’ Median Survival (Months)		
	(^Gene^/_LIFR_) > 1	(^Gene^/_LIFR_) < 1	*p-*Value
*YAP1*	28.7	85.6	0.0004 ***
*TAZ*	32.1	85.6	0.0085 *
*CTGF*	30.4	87	0.00058 **
*CYR61*	34.7	85.6	0.046 *
*AXL*	35.9	99.4	0.023 *
*AREG*	35.9	93.2	0.033 *

* *p* < 0.05, ** *p* < 0.005, *** *p* < 0.0005 versus untreated controls with log-rank test.

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
