# Peer review of "Leukaemia Inhibitory Factor (LIF) Inhibits Cancer Stem Cells Tumorigenic Properties through Hippo Kinases Activation in Gastric Cancer"

_cancers, 2020, doi:10.3390/cancers12082011_

Round 1

Reviewer 1 Report

The manuscript from Seeneevassen et al. outlines the regulation of cancer stem cells in gastric cancer and how a triggering LIFR/Hippo axis support a reduction in CSC levels and may benefit survival. The work is well laid out and interesting with the conclusions being mainly valid and supported by experimental data. There are suggestions for the introduction, results and figures as laid out below are intended to help improve the impact and context of the work.

Major comments.

The authors raise the review by Harvey et al. and indicating associations between the hippo pathway and cancer, this review only covered somatic mutations similar mimicking the drosophila genetics and unfortunately was not comprehensive. A key component in the Hippo pathway was missing from this analysis and has direct relevance to the study here, RASSFs are the most widely inactivated tumor suppressor in human malignancies  and associated with survival of every solid human tumour but often overlooked due to lack of phenotype in developmental systems. The authors should consider the following articles as supportive of the introduction to their study, namely as RASSF1A is a direct binder and activator of MST1/2.

  1. Shi et al. Association of RASSF1A promoter methylation with gastric cancer risk: a meta-analysis. Tumour Biol. 2014 Feb;35(2):943-8.
  2. Zhou et al. Polymorphism of A133S and promoter hypermethylation in Ras association domain family 1A gene (RASSF1A) is associated with risk of esophageal and gastric cardia cancers in Chinese population from high incidence area in northern China. BMC Cancer. 2013 May 25;13:259.

…and associates with stem cell behaviour

  1. Papaspyropoulos et al. RASSF1A uncouples Wnt from Hippo signalling and promotes YAP mediated differentiation via p73. Nat Commun. 2018 Jan 30;9(1):424.

Specific comments;

Fig 1A, B: The increase in pSTAT and transcriptional responses  are only apparent in AGS cells and barely visible in GC07 – I would remove MKN45 cells or put this in supplementary as an exception.

Fig 1D: It is not possible to read the axis and the gating looks slightly shifted between cell lines, are the authors sure AGS actually has an appreciable CD44 population? – it is clear from MKN45 FACS.

Fig 1E: Are there light bars hidden behind the darks bars for CD44+ pop?? I cannot see where the statistics are coming from.

What does 3 < n < 4 indicate?

Fig 2 – In general labelling is fuzzy and picture quality low.

Fig 2B, C:  MKN45 results are not convincing. Could you just leave out this cell line entirely from Fig 1and 2? As a suggesting you could remove these negative data and alter the flow to introduce Fig 2B, and 2C first without the MKN45 and then present the tumorsphere assays?

Fig 3 – Is it possible to Complete 3A with AGS and GC07?

Fig 4 – is nice data but again MKN45 data not strong or complete (without explanation) in Fig 4A and detracts from AGS results. I would complete the panel of WBs for MKN45 and pu tin supplementary also include MST1/2 and pMST1/2 at least for AGS.

You could also consider WB for Sav1 and RASSF1 and the bone fide activators of MST that could potentially be downstream of LIFR.

Fig 4C is fuzzy.

Fig 5A – again MST1/2 WB as this is the target,

Fig 6 is a nice piece of data but it is difficult to see, it would be to your avantage to display this better or at least make it legible.

Author Response

Dear reviewer,

We thank you very much for your comments that led to a significant improvment of the quality of our manuscript.

Please find in the attached document our point by point responses to your comments.

Reviewer 2 Report

The authors examine the effect of LIF on the cancer stem cell population in gastric cancer and find that the Hippo pathway is important to maintain CSC properties. In a complimentary publication, the same group used similar methods to determine that inhibition of YAP/TAZ with verteporfin targets the CSC population of GC. These conclusions, while not completely overlapping, are very similar.

Beyond the question of novelty, this is a clearly and nicely written article. The data, by and large, is convincing and solid. The conclusions are persuasive and potentially clinically relevant.

Major comments:

  1. The study relies solely on the effect of kinase inhibitors which are notorious for their off-target activity. Major conclusions should be backed up with knock-down experiments.
  2. It would have been interesting (and more clinically relevant) to examine the effect of combined LIF and verteporfin treatment. Is all the effect of LIF via YAP/TAZ inhibition, or is there an additive effect in combined treatment?
  3. Similarly, does LIF act in parallel on the JAK/STAT and Hippo pathways (as depicted in authors' graphic summary), or is there a more intimate cross-talk between the two pathways? To determine this, YAP/TAZ targets should be measured in response to Ruxo; and JAK/STAT targets (or activity) should be measured in response to XMU treatment.
  4. If LIF treatment is the focus of the study, gene expression should have been profiled in LIF treated vs nontreated CSCs. Instead, the article starts off with weak data relating to the JAK/STAT pathway. Clear activation of STAT3 in response to LIF treatment (Fig 1A) occurs only in AGS cells. Moreover, it appears that negative regulators of JAK/STAT are more expressed than positive JAK/STAT regulators (Fig 1C), suggesting that the JAK/STAT pathway might be inactivated in CSCs.
  5. The GP190 data should be presented (Lines 184-185). This is important complimentary data to the (weak) LIFR data, in order to justify the statement that the LIF response is primarily attributed to CSC cell population (lines 191-193). [Although significant, the difference in LIFR protein expression in AGS cells (Fig 1E) is very mild and on the RNA level there is no difference at all (Fig 1C). STAT target gene expression should have been profiled in LIF treated vs nontreated CSCs.] Similarly, K-M plots for GP190 could be presented. What is the relation between LIFR and GP190 and patient survival?
  6. What is the relative portion of CD24+CD44+ CSCs within the GC population? According to Fig 1E, it looks like it's almost 50% of the total population. Is that logical?

Similarly, is it unexpected that there are not two distinct subpopulations of CD44+ vs CD44low cells? This becomes important conceptually. Is there a defined CSC population, or is there a spectrum of expression of CD44? The former may indicate that targeting a plastic CSC population might be quite difficult, particularly in the PDX and AGS models, in which most of the data rests and where CD44 expression appears as a continuum (Fig 1 and 2).

Minor comments:

  1. Fig 1E: "$" symbol usage should be explained in figure legend
  2. Fig 3A. In the non-treated population, most CD44+ cells are Hoechst positive. Is this not counter intuitive?
  3. The tumorshpere assays are very nice. Is the decrease in sphere producing CSCs due to CSC death, or inhibition of self-renewal capacity?
  4. Fig 4A. The blot for AGS cells is very strange. All antibodies (including even GAPDH) show an uncharacteristic doublet.

The authors should indicate the relevant band. Particularly because everything is based on the difference between lane 1 and 2, where the top "GAPDH" band also seems to increase.

  1. General comment: figure panels should be arranged in alphabetical order. In the current state, several of the panels are awkwardly arranged.
  2. The graphic representation of Fig 4B (presented in Fig 4C) do not seem to correspond. (See specifically AGS cells at 5hs)
  3. The lanes of the Westerns presented in Fig 5A appear to be cropped from separate gels. If so, cuts should be clearly marked by placement in separate frames.
  4. In AGS cells, treatment with XMU seems to primarily affect LATS2 protein levels (as opposed to a specific decline in phosphorylation of LATS).
  5. Relating to Fig 4 and 5, authors should present also total LATS1 protein levels.
  6. In human tumor data, do those tumors with low LIFR and bad prognosis also have high expression of YAP/TAZ targets? In other words, might these be the patients that would benefit from combined LIF and verteporfin treatment?

Author Response

Dear reviewer,

We thank you very much for your comments that led to a significant improvement of the quality of our manuscript.

Please find below our point by point responses to your comments.

Christine Varon, corresponding author

Reviewer 3 Report

The authors evaluated the role of Leukaemia Inhibitory Factor (LIF) in the gastric cancer (GC) in this manuscript. The authors presented that LIF acted with anti‐tumorigenic potential in GC cell lines and the Hippo pathway was at the basis of LIF‐induced anti‐cancer stem cells (CSCs) effects in GC, suggesting that LIF could have anti‐CSC therapeutic potential in combination with conventional chemotherapy. Lastly, the authors presented that low expression of LIFR was observed in GC tissue and low LIF/LIFR was associated to poor survival in diffuse type GC, suggesting that LIF/LIFR signaling represents a possible prognosis marker and therapeutic option in the management of GC. Some major and minor points described below should be addressed.

  1. The authors need to describe the detail information of cancer cell lines used in this study.
  2. The authors also need to describe the detail information of PDX-derived cell lines while they had specific reference for lines in the manuscript.
  3. While both AGS and MKN45 cells are established from intestinal type GC, the results of the experiments seem to have big differences between these two lines. The authors need to explain this point.
  4. The authors showed that LIF could inhibit CSCs tumorigenic properties through Hippo kinases activation in GC in this manuscript. Now I wonder how LIF could work on EMT and/or invasion potential, as CSCs are also related with these phenomena.
  5. The authors showed that LIF could inhibit CSCs tumorigenic properties through Hippo kinases activation in AGS and MKN45 cells (intestinal type GC cells), while high LIF was related with poor prognosis of patients with intestinal type GC. The authors need to explain this discrepancy. The authors described “This difference in prognosis of GC patients again shows how LIF is pleiotropic and can lead to different fates depending on the type of cancer.” However, I am afraid that this explanation is not sufficient.
  6. Figure 3B: The authors described “when LIF was added in combination with DOX and 5‐FU, thenumber of cells forming spheres in vitro decreased significantly compared to chemotherapy drugs”. However, GC07 cells seemed to work differently.
  7. Figure 4A: An increase in LATS1/2-mediated phosphorylation of YAP on Ser127 in a time-dependent manner was not observed in MKN45 cells.

Author Response

Dear reviewer,

We thank you very much for your comments that led to a significant improvement of the quality of our manuscript.

Please find below our point-by-point responses to your comments.

Christine Varon, corresponding author

Reviewer 4 Report

Dear Editor-in-chief

Cancers

Regarding manuscript Review cancers-828586, entitled `` Leukaemia Inhibitory Factor (LIF) inhibits Cancer Stem Cells tumorigenic properties through Hippo kinases activation in Gastric Cancer`` by Lornella Seeneevassen et al.

Current paper described the Leukaemia Inhibitory Factor (LIF) on cancer stem cells (CSC) properties in gastric cancer (GC) cell lines and patient‐derived xenograft (PDX) cells. Authors have shown that Hippo kinase inhibitor (XMU‐MP‐1) and/or JAK1 inhibitor (Ruxolitinib), Data showed that LIF reduced tumorigenic and chemo resistant CSCs in GC cell lines and patient‐derived xenograft cells. XMU‐MP‐1 inhibitor has inhibited LIF and its anti-tumor effects, however Ruxolitinib showed anti-tumor effects.

Paper has been written well with good information. Before considering the manuscript for publication, authors are suggested to address the following issues:

Major comments:

  • Do you think that two GC cell lines, AGS and MKN45 GC are enough to reach the conclusion? Justify it in manuscript.
  • How you used a negative control cell line to check the specificity of your data?
  • Figure 1A: For cell line MKN 45, it seems there is no significant differences at time 0 to 48h for STAT3 phosphorylation. Even it is clear that at time points 5 and 24 h it decreased. As well as GC07 PDX.
  • Why authors did not check the phosphorylation of JAK after addition of recombinant LIF?
  • Using an inhibitor of STAT3 or JAK is necessary to show the inactivation of the pathway as the system control.
  • Figure 1b: Increase in expression of JAK1/STAT3 target genes were not constant in some cell lines or after upregulation were downregulated later. Justify this and describe the reasons. High variation for genes and different cell lines I obvious, why?
  • Authors have not shown similar data for PDX cells in figure 1D. What is the reason?
  • Due the above questions, is LIF treatment an appropriate strategy, as claimed in line 194-197?
  • Figure 2A: 10ng/ml of LIF has been shown only for the AGS cell line. What is the status for other cell line at this concentration of LIF?
  • Figure 2B: Have you checked the activity of CD44 and ALDH activity or % of positive cells. If it is wrong, correct the statements.
  • The last Figure of 2B, lower panel, % of ALDH belongs to which cell line?

Minor comments:

  • Uniform abbreviations in manuscript. E.g. some parts it is written 1 hour and other places 1h etc…
  • Is it common to see CSCs population in different cell lines with different origin or treatment with soluble factors is necessary to differentiate them, as shown in Figure 1D ???
  • What is the reason for using 2 different loading controls in WB experiments?

Author Response

(The authors gave the same response as above.)

Round 2

Reviewer 3 Report

The authors have responded to almost all of the reviewers' comments. Additional detailed information provided in the revised manscript and authors comments made the paper more clearly.

Figure 3B: In GC07 cells, anti-tumoursphere forming effect of  DOX+LIF was less than that of LIF alone. Please explain this point.

Reviewer 4 Report

Dera Editor-in-chief

Cancers

Authors have properly responded the questions